# Correlation of sLOX-1 Levels and MR Characteristics of Culprit Plaques in Intracranial Arteries with Stroke Recurrence

**DOI:** 10.3390/diagnostics13040804

**Published:** 2023-02-20

**Authors:** Kaixuan Ren, Huayun Jiang, Tiantian Li, Chengqun Qian, Li Zhu, Tianle Wang

**Affiliations:** 1The Second Affiliated Hospital of Nantong University, Nantong 226000, China; 2Affiliated Hospital of Yangzhou University, Yangzhou 225000, China

**Keywords:** atherosclerosis, stroke recurrence, vessel wall magnetic resonance imaging, soluble lectin-like oxidised low-density lipoprotein receptor 1

## Abstract

(1) Background: Symptomatic intracranial artery atherosclerosis (sICAS) is an important cause of acute ischaemic stroke (AIS) and is associated with a high risk of stroke recurrence. High-resolution magnetic resonance vessel wall imaging (HR-MR-VWI) is an effective method for evaluating atherosclerotic plaque characteristics. Soluble lectin-like oxidised low-density lipoprotein receptor-1 (sLOX-1) is closely associated with plaque formation and rupture. We aim to explore the correlation between sLOX-1 levels and culprit plaque characteristics, based on HR-MR-VWI, with stroke recurrence in patients with sICAS. (2) Methods: A total of 199 patients with sICAS underwent HR-MR-VWI between June 2020 and June 2021 in our hospital. The culprit vessel and plaque characteristics were assessed according to HR-MR-VWI, and sLOX-1 levels were measured by ELISA (enzyme linked immunosorbent assay). Outpatient follow-up was performed 3, 6, 9, and 12 months after discharge. (3) Results: sLOX-1 levels were significantly higher in the recurrence group than in the non-recurrence group (*p* < 0.001). The culprit plaque thickness, degree of stenosis and plaque burden were higher in the recurrence group than in the non-recurrence group (*p* = 0.003, *p* = 0.014 and *p* = 0.010, respectively). The incidence of hyperintensity on T1WI, positive remodelling and significant enhancement (*p* < 0.001, *p* = 0.003 and *p* = 0.027, respectively) was higher in the recurrence group than in the non-recurrence group. Kaplan–Meier curves showed that patients with sLOX-1 levels > 912.19 pg/mL and hyperintensity on T1WI in the culprit plaque had a higher risk of stroke recurrence (both *p* < 0.001). Multivariate Cox regression analysis showed that sLOX-1 > 912.19 pg/mL (HR = 2.583, 95%CI 1.142, 5.846, *p* = 0.023) and hyperintensity on T1WI in the culprit plaque (HR = 2.632, 95% CI 1.197, 5.790, *p* = 0.016) were independent risk factors for stroke recurrence. sLOX-1 levels were significantly associated with the culprit plaque thickness (r = 0.162, *p* = 0.022), degree of stenosis (r = 0.217, *p* = 0.002), plaque burden (r = 0.183, *p* = 0.010), hyperintensity on T1WI (F = 14.501, *p* < 0.001), positive remodelling (F = 9.602, *p* < 0.001), and significant enhancement (F = 7.684, *p* < 0.001) (4) Conclusions: sLOX-1 levels were associated with vulnerability of the culprit plaque and can be used as a supplement to HR-MR-VWI to predict stroke recurrence.

## 1. Introduction

Recurrent stroke exposes the brain tissue to renewed hypoperfusion and oxidative stress, further exacerbating neurological damage and significantly increasing the rate of severe disability and mortality [1,2]. Symptomatic intracranial atherosclerotic stenosis (sICAS) is one of the most common causes of ischaemic stroke, including acute ischaemic stroke (AIS) and transient ischaemic attack (TIA), and it is associated with a higher risk of stroke recurrence [3]. Current practice guidelines determine treatment strategies based on the degree of stenosis [4]. However, recent studies have reported that the impact on stroke is not due to luminal stenosis caused by plaque but rather due to the nature of the culprit plaque itself [5]. High-resolution magnetic resonance vessel wall imaging (HR-MR-VWI) has been recently an important tool for assessing atherosclerosis. Previous studies on the internal carotid artery or the middle cerebral artery showed an association between the positive wall remodelling, hyperintensity on the T1-weighted imaging (T1WI), plaque enhancement and stroke recurrence [6].

Lectin-like oxidised low-density lipoprotein receptor 1 (LOX-1) is the major receptor of oxidised low-density lipoprotein (ox-LDL). Soluble lectin-like oxidised low-density lipoprotein receptor 1 (sLOX-1) is a soluble scavenger receptor released by protease hydrolysis of LOX-1 on the cell surface, and therefore, sLOX-1 levels reflect the expression level of LOX-1 [7]. Under physiological conditions, LOX-1 is mainly expressed on vascular endothelial cells; however, in advanced stages of atherosclerotic lesions, it can be extended to smooth muscle cells and macrophages for expression [8], which affects the stability of atherosclerotic plaques by inducing the release of matrix metalloproteinases from macrophages [9]. An increasing number of basic and clinical studies have demonstrated a correlation between sLOX-1 levels and atherosclerotic-like diseases [10]; however, the correlation between sLOX-1 levels and the risk of stroke recurrence needs to be further investigated.

This study aimed to investigate the correlation between culprit plaque characteristics and sLOX-1 levels, and stroke recurrence in patients with sICAS, as well as to conduct a preliminary investigation of the relationship between sLOX-1 levels and vulnerability of the culprit plaque.

## 2. Materials and Methods

### 2.1. Clinical Data and Biochemical Indicators

This was a single-centre retrospective cohort study. Retrospective analysis of 256 patients with suspected sICAS who underwent HR-VWI in our hospital between June 2020 and June 2021 was performed. All patients were classified as having a large atherosclerotic stroke of trial of ORG 10172 in acute stroke treatment (TOAST) type or indicated that the ischaemic event was caused by intracranial atherosclerosis. The inclusion criteria were as follows: (1) complete baseline demographic data and atherosclerotic risk factors data; (2) HR-VWI examination was performed within one week of onset, and all culprit vessels had plaque formation. The exclusion criteria were as follows: (1) non-atherosclerotic vascular disease, such as vascular malformation or intracranial aneurysm (*n* = 9); (2) received endovascular therapy (*n* = 12); (3) ipsilateral extracranial artery stenosis ≥ 50% (*n* = 8); (4) combined with potential cardiogenic embolic factors (e.g., atrial fibrillation) (*n* = 12); and (5) poor image quality (*n* = 16). A total of 199 patients were included in the study, including 148 with AIS and 51 with TIA. The patient selection strategy is shown in Figure 1. Each patient or their relatives signed the study consent form before inclusion in the study, which was approved by the Ethics Committee of the Second Affiliated Hospital of Nantong University (No. 2016YXJS010).

We collected demographic data and atherosclerotic risk factors from all patients within 24 h of admission. All patients were started on dual antiplatelet treatment with aspirin (150–300 mg/day) and clopidogrel (75 mg/day, first 300 mg) within 24 h after admission, and they adhered well to regular medication during the follow-up period. Outpatient follow-up was performed 3, 6, 9, and 12 months after discharge. Stroke recurrence was defined as the presence of a new acute infarct focus in the same vascular supply area on diffusion-weighted imaging (DWI) (*n* = 30). When no imaging was available for the suspected recurrent event, the follow-up was based on the characteristics of the new neurological deficit symptoms (National Institute of Health stroke scale (NIHSS) increase > 4 points) and duration (>24 h) to determine the occurrence of the outcome event (*n* = 11) [11]. The follow-up time was defined as during the time of diagnosis to the endpoint events or to the most recent follow-up if no event occurred. The “last observation carried for-ward” protocol was followed for incomplete follow-up data.

### 2.2. HR-VWI Examination

HR-MR-VWI images were acquired using a Siemens Verio 3.0 T MR imaging system (Siemens, Erlangen, Germany). In addition to the conventional MRI and MRA scanning sequence, three-dimensional T1 sampling perfection with application optimised contrasts using different flip angle evolution (3D-T1-SPACE) sequence was scanned before and after injecting the contrast agent with the following parameters: repetition time (TR) = 700 ms, echo time (TE) = 12 ms, field of view (FOV) = 200 × 200 × 40 mm^3^, and voxel size = 0.8 × 0.8 × 0.7 mm^3^. Gadoteric acid was used as the contrast agent at a dose of 0.1 mmol/kg, and the enhanced images were collected at 8 min after injection. The scanning range included the complete presentation of the intracranial segment of the internal carotid artery, M1 and M2 segments of the middle cerebral artery, A1 and A2 segments of the anterior cerebral artery, P1 and P2 segments of the posterior cerebral artery, V4 segment of the vertebral artery, and the basilar artery.

### 2.3. Image Analysis

The culprit vessel was located according to the acute infarct focus or neurological deficit sign on DWI, and the presence or absence of stenosis in the culprit vessel was determined on TOF-MRA images. The location of the culprit plaque was determined on a 3D-T1-SPACE sequence combined with sagittal, coronal, and axial images; that is, the limited or diffuse thickening of the vessel wall was greater than 50% of the adjacent vessel wall [12]. When multiple plaques were present in the culprit vessel, the one causing the most severe stenosis was defined as the culprit plaque. A Siemens workstation (MMWP Siemens Erlangen) was used for image post-processing. All images were analysed by a second-year neuroimaging graduate student and a neuroradiologist with five years of experience. A third neuroradiologist with 10 years of experience participated in the analysis when there was a disagreement, and agreement was reached after consultation.

The maximum thickness of the vessel wall at the culprit plaque was defined as the plaque thickness, whereas the remaining lumen area was considered the culprit plaque. The reference site was selected according to the Warfarin–Aspirin Symptomatic Intracranial Disease method [13].

The degree of stenosis and plaque burden were defined using the following formulae [14,15]:degree of stenosis = (1 − lumen diameter at the culprit plaque/lumen diameter at the reference site) × 100%;
plaque burden = (1 − lumen area/vessel area) × 100%.

The remodelling index was calculated as the ratio of the area of the vessel zone at the culprit plaque to the vessel range at the reference site. A remodelling index higher than 1.05 was considered positive, whereas an index within 0.95–1.05 was designated as no remodelling. Finally, values smaller than 0.95 were defined as negative remodelling [16]. Another classification in this report concerned wall thickening. Centripetal thickening was defined as involvement of more than half of the arterial wall circumference. Less than 50% centripetal thickening was defined as eccentric. When the T1WI signal intensity of the culprit plaque was 150% higher than that of the ipsilateral masseter, it was defined as hyperintensity on T1WI. Significant enhancement was defined as a signal intensity close to or higher than that of the pituitary stalk [12]. Case presentations are shown in Figure 2.

We selected randomly 20 patients to test the intra-observer and inter-observer reproducibility in identifying the presence of hyperintensity on T1WI, significantly enhanced, the positive remodelling and measuring the plaque thickness, the remaining canal area, the degree of stenosis and the plaque burden were evaluated. A time interval of 1 month was set for determining the intra-observer reproducibility to minimize the bias of memory.

### 2.4. Statistical Analysis

The software SPSS 21.0 (IBM Corp., Armonk, NY, USA) was used for analysis, and the Kolmogorov–Smirnov test was used to determine whether the data conformed to a normal distribution. Normally distributed variables were expressed as mean ± standard deviation and *t*-test was performed; skewed variables were expressed as M (Q25, Q75) and the Mann–Whitney U test was performed; count data were expressed as frequency (percentage) and the chi-square test was performed. Variables with *p*-values <0.1 in the univariate analysis were included in the Cox regression analysis to further screen for risk factors independently associated with stroke recurrence. The subject working characteristic curves for predicting stroke recurrence at the sLOX-1 level in 199 patients were plotted, the area under the subject working characteristic curve (AUC) as well as the sensitivity and the specificity at the optimal cutoff value were calculated, and the Delong test was used to compare the differences between the AUC values. Kaplan–Meier survival curves were plotted using MedCalc 19.0. Pearson’s correlation was performed between sLOX-1 levels and culprit plaque quantitative characteristics. Differences in the qualitative characteristics of culprit plaques between quartile groups at the sLOX-1 level were assessed using grouped multiple sample ANOVA. A Kappa consistency test was used to evaluate the consistency between observation and internal repeatability of qualitative data, and intra-class correlation coefficients (ICC) were used to evaluate the consistency between observer and internal repeatability of quantitative data, and the Kappa value and ICC value greater than 0.75 were considered to be reproducible. All reported *p*-values were two-tailed, *p*-values < 0.05 were considered statistically significant.

## 3. Results

### 3.1. Comparison of Baseline Demographic Data and Differences in HR-VWI Characteristics between the Recurrence Group and the Non-Recurrence Group

A total of 199 patients were included in this study, and 41 developed stroke recurrence during the 12-month follow-up period. Overall, 30 patients had a new acute infarct focus in the same vascular supply area on DWI and 11 patients were diagnosed as new neurological deficit symptoms. The mean age of the patients was 67 ± 10 years in the recurrence group and 64 ± 13 years in the non-recurrence group. The proportion of men in the recurrence group, history of smoking, and history of diabetes were higher than those in the non-recurrence group, but the difference was not statistically significant. Serum glycated haemoglobin, triglyceride, total cholesterol, apolipoprotein B, LDL, homocysteine, and cystatin C levels were higher in the recurrence group than in the non-recurrence group, while serum apolipoprotein A and high-density lipoprotein (HDL) levels were higher in the non-recurrence group than that in the recurrence group, but the differences were not statistically significant. Serum LOX-1 levels were significantly higher in the recurrence group than in the non-recurrence group, and the difference was statistically significant (t = −4.29, *p* < 0.001) (Table 1).In the recurrence group, 21 patients (51.2%) had culprit plaques in the posterior circulation, which was higher than that in the non-recurrence group (44.3%). The lumen area at the culprit plaque in the recurrence group was smaller than that in the non-recurrence group, but the differences were not statistically significant. As for the quantitative indices, the culprit plaque thickness (t = −2.19, *p* = 0.003), stenosis (t = −2.48, *p* = 0.014), and plaque burden (t = −2.57, *p* = 0.010) were higher in the recurrence group than in the non-recurrence group, and the differences were statistically significant. Regarding the qualitative indices, the incidence of hyperintensity on T1WI (χ^2^ = 21.31, *p* < 0.001), positive remodelling (χ^2^ = 9.33, *p* = 0.003), and significant enhancement (χ^2^ = 5.83, *p* = 0.027) in the culprit plaques were higher in the recurrence group than that in the non-recurrence group, and the difference was statistically significant (Table 2).For the intra-observer agreement in the identification of the presence of hyperintensity on T1WI, which was significantly enhanced, the positive remodelling and the kappa value were 1.00, 0.96, and 0.93 (all *p*  <  0.001), respectively. For the inter-observer agreement, the values of the same parameters were 1.00, 0.92 and 0.89 (all *p*  <  0.001), respectively. The intra-observer ICC of plaque thickness, remaining lumen area, degree of stenosis and plaque burden were 0.90 (95% CI, 0.76–0.95, *p*  <  0.001), 0.95 (95% CI, 0.91–0.98, *p*  <  0.001), 0.88 (95% CI, 0.74–0.95, *p*  <  0.001) and 0.80 (95% CI, 0.63–0.90, *p*  <  0.001), respectively. The inter-observer ICC values for the above-described parameters were as follows: 0.94 (95% CI, 0.84–0.97, *p*  <  0.001), 0.96 (95% CI, 0.93–0.99, *p*  <  0.001), 0.87 (95% CI, 0.75–0.93, *p*  <  0.001) and 0.86 (95% CI, 0.75–0.94, *p*  <  0.001).

### 3.2. Independent Risk Factor Analysis for Stroke Recurrence and Survival Curve Analysis

All indicators (*p* < 0.1) in the univariate analysis were included in the multivariate Cox regression analysis, which showed that sLOX-1 levels (HR = 1.001, 95% CI 1.000, 1.002, *p* = 0.002) and hyperintensity on T1WI in the culprit plaque (HR = 2.326, 95% CI 1.034, 5.231, *p* = 0.041) were independent risk factors for stroke recurrence (Table 3). We plotted ROC curves of sLOX-1 levels and hyperintensity on T1WI to predict stroke recurrence (Figure 3). Based on the ROC curve analysis, the optimal cutoff value of sLOX-1 to predict stroke recurrence was 912.19 pg/mL with an AUC value of 0.707, sensitivity of 80.49%, and specificity of 52.53%; the AUC value of hyperintensity on T1WI to predict stroke recurrence was 0.724, with a sensitivity of 70.73% and specificity of 74.05%. However, the Delong test showed that the differences between the AUC values was not statistically significant (Z = 0.315, *p* = 0.753). Kaplan–Meier survival curves showed that patients with sLOX-1 levels >912.19 pg/mL (*p* < 0.001) and hyperintensity on T1WI in the culprit plaque (*p* < 0.001) had a higher risk of stroke recurrence at follow-up (Figure 4). For further study, we converted sLOX-1 levels into a binary variable (sLOX-1 > 912.19 pg/mL was defined as 1, and sLOX-1 ≤ 912.19 pg/mL was defined as 0). Multivariate Cox regression analysis showed that sLOX-1 > 912.19 pg/mL (HR = 2.583, 95%CI 1.142, 5.846, *p* = 0.023) and hyperintensity on T1WI in the culprit plaque (HR = 2.632, 95% CI 1.197, 5.790, *p* = 0.016) were independent risk factors for stroke recurrence (Table 4).

### 3.3. Relationship between sLOX-1 Levels and Culprit Plaque Characteristics

Pearson’s correlation analysis showed that sLOX-1 levels were significantly correlated with the culprit plaque thickness (r = 0.162, *p* = 0.022), degree of stenosis (r = 0.217, *p* = 0.002), and plaque burden (r = 0.183, *p* = 0.010) (Figure 5). To further explore the correlation between sLOX-1 levels and culprit plaque qualitative characteristics, we observed quartiles of sLOX-1 levels. The results showed that the proportion of hyperintensity on T1WI, positive remodelling and significantly enhanced in each quartile group showed a gradient increasing trend with the increase in sLOX-1 levels. Analysis of variance of the multiple grouped samples showed statistically significant differences in the incidence of hyperintensity on T1WI (F = 14.501, *p* < 0.001), positive remodelling (F = 9.602, *p* < 0.001), and significant enhancement (F = 7.684, *p* < 0.001) among the quartile groups (Table 5).

## 4. Discussion

A total of 199 patients were included in this study, and 41 developed stroke recurrence during the 12-month follow-up period. sLOX-1 levels were significantly higher in the recurrence group than in the non-recurrence group, and the difference was statistically significant. For quantitative indices based on HR-MR-VWI, the culprit plaque thickness, stenosis, and plaque burden were higher in the recurrence group than in the non-recurrence group. As for the qualitative indices, the incidence of hyperintensity on T1WI, positive remodelling, and significant enhancement of the culprit plaque were higher in the recurrence group than that in the non-recurrence group. Multivariate Cox regression analysis showed that sLOX-1 levels and hyperintensity on T1WI of the culprit plaque were independent predictors of stroke recurrence in patients with sICAS.

In our study, hyperintensity on T1WI of the culprit plaque was an independent risk factor for stroke recurrence. Hyperintensity on T1WI is often indicative of subacute bleeding within atherosclerotic plaques associated with plaque progression, lipid core formation, and instability, and it is an important marker for distinguishing vulnerable plaques from stable plaques [17,18].

Several recent studies have concluded [19,20] that the degree of plaque enhancement is correlated with stroke recurrence. Plaque enhancement is usually considered a characteristic sign of plaque progression or instability and is an important predictor of stroke recurrence [21]. Plaque enhancement may be due to the combined effects of massive neovascularisation caused by inflammation, vessel wall thinning, and inflammatory cell infiltration [22]. Our study showed that culprit plaque enhancement was higher in the recurrence group than that in the non-recurrence group, but it was not an independent risk factor for stroke recurrence within one year. The reason for this may be our failure to quantify the degree of enhancement and relatively short follow-up period [23].

Several other follow-up studies based on HR-MR-VWI have found that plaque burden [24] and its progression over time [25] are associated with stroke recurrence. Our study quantified the plaque burden of the culprit plaque but did not obtain similar results; this may be due to our short follow-up time. In addition, the luminal stenosis of intracranial arteries based on conventional imaging modalities has been considered a high-risk factor for stroke recurrence [26]; however, the results of this study suggest that luminal stenosis is not associated with stroke recurrence, which is probably because the stenosis our study is obtained by objective calculation, and it is the actual stenosis after the influence of arterial remodelling [27].

We followed the enrolled patients for a mean of 1 year and found that sLOX-1 levels could be used as an independent predictor of stroke recurrence. sLOX-1 predicted stroke recurrence with a threshold of 912.19 pg/mL, an AUC value of 0.707, a sensitivity of 80.49%, and a specificity of 52.53%. sLOX-1 is the major receptor for ox-LDL that mediates the development of atherosclerosis. It is widely expressed on the surface of atherosclerosis-associated cells, and it is detected in the circulatory system by proteolytic cleavage into a soluble form (sLOX-1) [28]. There is growing evidence that sLOX-1 is involved in major steps in the pathogenesis of atherosclerosis [29].

Our results showed that sLOX-1 levels were significantly associated with the culprit plaque thickness, degree of stenosis, plaque burden, hyperintensity on T1WI, positive remodelling and significant enhancement. sLOX-1 levels were significantly correlated with vulnerable plaque properties, such as culprit plaque thickness, plaque hyperintensity on T1WI, and significant enhancement. When LDL enters the extracellular matrix within the vessel wall, oxidative modification can occur to form ox-LDL. The process of ox-LDL uptake by inflammatory cells and removal of excess lipids can then form plaques that gradually evolve into vulnerable plaques, which are characterised by massive inflammatory cell infiltration and a thin fibrous cap covering the eccentric necrotic lipid core [30]. sLOX-1 increases intra-plaque through the activation of matrix metalloproteinase degradation of fibrous tissue, which is the key process by which plaques become fragile and undergo rupture [31]. Therefore, sLOX-1 levels can, to some extent, reflect the vulnerable nature of plaques. sLOX-1’s role in the development of vulnerable plaques and the relationship between sLOX-1 levels and MR characteristics of vulnerable plaques need to be further confirmed by more basic studies and clinical cohort studies in the future.

Although the analysis of culprit plaque vulnerability based on HR-MR-VWI can effectively predict the risk of stroke recurrence, it is not applicable to patients with critical conditions or contraindications to MRI, and the relatively long examination time of HR-MR-VWI is a limitation for routine clinical implementation. Our study found no significant difference between the predictive value of sLOX-1 for stroke recurrence and that of hyperintensity on T1WI in culprit plaques. sLOX-1, a more easily available biomarker, is expected to be a complementary tool to HR-MR-VWI and plays an important clinical value for the prediction of stroke recurrence.

This study had some limitations. First, it was a single-centre study with a small sample size. Given the long duration of HR-MR-VWI, patients with more stable disease and better cooperation were more likely to be included in this study; therefore, the probability of selection bias was higher. Multivariate analyses were performed to make the results as objective as possible. Second, regarding the measurement of sLOX-1 levels, this study did not consider changes in sLOX-1 levels during different periods of stroke, and dynamic monitoring of sLOX-1 levels is required in the future. Third, other imaging indicators associated with recurrent stroke, including plaque surface irregularity and hemodynamic impairments, were not included in the study. The AUC and specificity of sLOX-1 levels predicting stroke recurrence were relative low, and further validation is needed in larger multi-centre prospective studies. In addition, with the development of an automated method for HR-MR-VWI, including vessel centreline tracking, vessel straightening and reformation, vessel wall segmentation based on convoluted neural networks (CNNs), and morphological measurement, it is expected to facilitate large-scale arterial vessel wall morphological quantification [32].

## 5. Conclusions

The sLOX-1 level and hyperintensity on T1WI of culprit plaques are independent risk factors for stroke recurrence, and sLOX-1 levels are significantly correlated with vulnerable plaque properties such as culprit plaque thickness, plaque hyperintensity on T1WI, and significant enhancement. The sLOX-1 level can be an important complementary tool for assessing the degree of intracranial atherosclerosis in patients with sICAS, and it may be an important guide for risk stratification management of stroke patients and the development of individualised treatment plans.

## Figures and Tables

**Figure 1 diagnostics-13-00804-f001:**
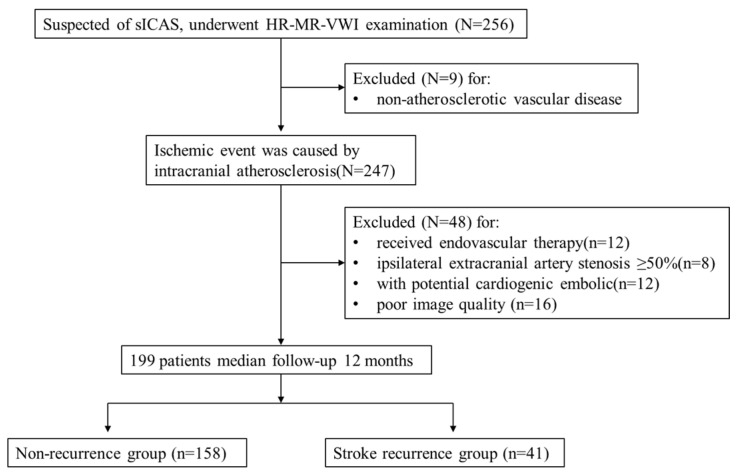
Flow chart showing the selection of patients.

**Figure 2 diagnostics-13-00804-f002:**
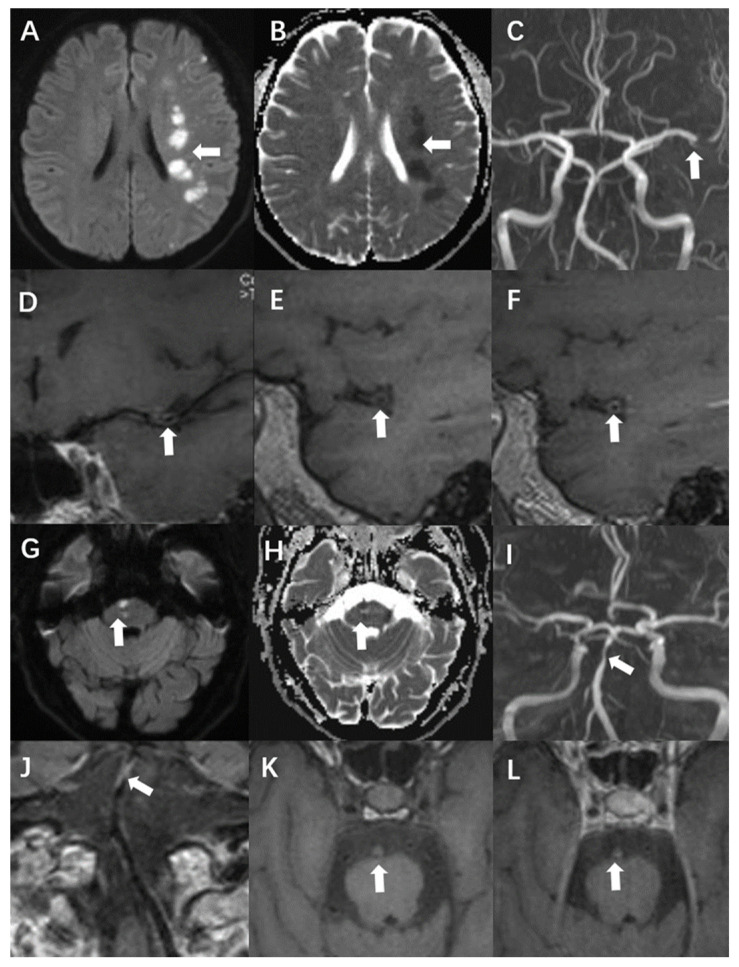
A 54-year-old man presented with acute infarction in the left cerebral hemisphere (**A**–**F**). No recurrence was observed during follow-up, and the sLOX-1 level was 214.46 pg/mL. (**A**,**B**): High DWI signal (white arrow) and low apparent diffusion coefficient (ADC) value (white arrow) in the left cerebral hemisphere. (**C**): Localised stenosis of the distal segment of the left middle cerebral artery (white arrow). (**D**): Localised thickening and enhancement of the distal wall of the left middle cerebral artery (white arrow). (**E**): Culprit plaque 3D-T1-space plain scan showed isosignal (white arrow). (**F**): There was no significant enhancement after the enhancement of culprit plaque (white arrow). A 69-year-old man with acute pontine infarction was diagnosed with recurrent stroke 6 months after discharge, with an sLOX-1 level of 1429.44 pg/mL (**G**–**L**). (**G**,**H**): DWI in the right pons is high (white arrow), and the ADC value is decreased (white arrow). (**I**): Basilar artery localised stenosis (white arrow). (**J**): Basilar artery wall localised thickening with enhancement (white arrow). (**K**): Hyperintensity was observed on the 3D-T1-space plain scan of the culprit plaque (white arrow). (**L**): There was no significant enhancement after enhancement of the culprit plaque (white arrow).

**Figure 3 diagnostics-13-00804-f003:**
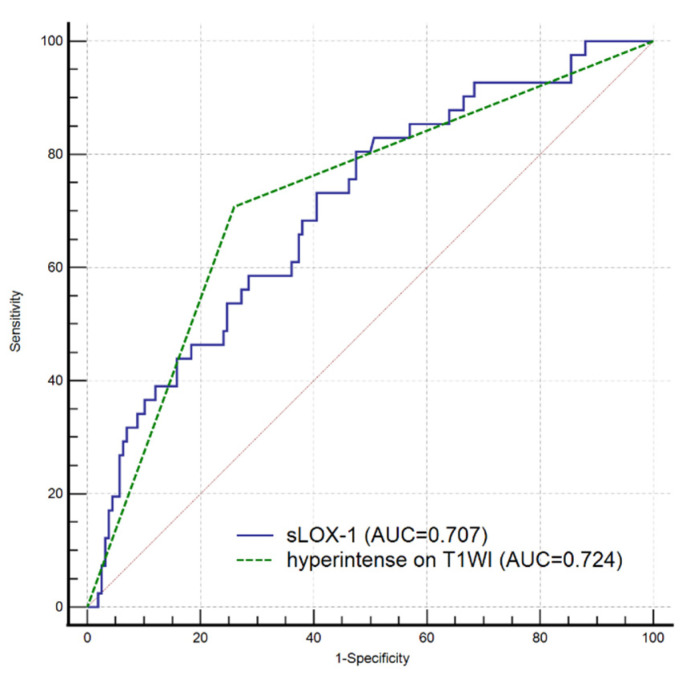
ROC curves of sLOX-1 level and hyperintensity on T1WI to predict stroke recurrence. The optimal cutoff value of sLOX-1 to predict stroke recurrence was 912.19 pg/mL with an AUC value of 0.707, sensitivity of 80.49%, and specificity of 52.53%; the AUC value of hyperintensity on T1WI to predict stroke recurrence was 0.724, sensitivity of 70.73% and 74.05% specificity. However, the difference was not statistically significant (Z = 0.315, *p* = 0.753).

**Figure 4 diagnostics-13-00804-f004:**
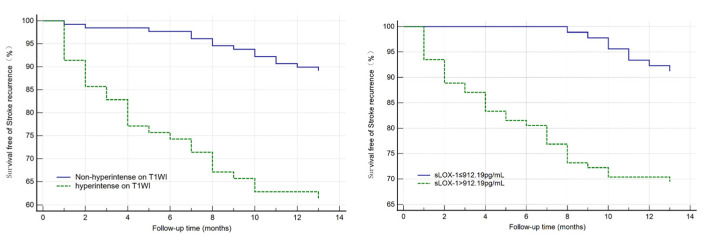
Kaplan–Meier survival curve in patients with stroke recurrence. Patients with sLOX-1 levels > 912.19 pg/mL (*p* < 0.001) and hyperintensity on T1WI in the culprit plaque (*p* < 0.001) had a higher risk of stroke recurrence at follow-up.

**Figure 5 diagnostics-13-00804-f005:**
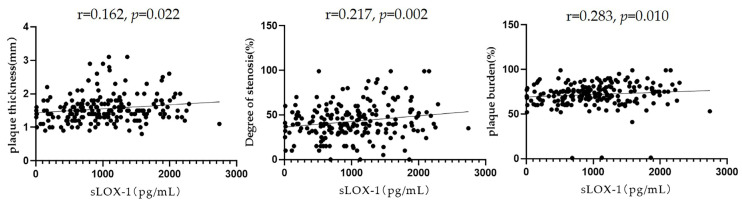
Relationship between sLOX-1 levels and culprit plaque quantitative characteristics. sLOX-1 levels were significantly correlated with the culprit plaque thickness (r = 0.162, *p* = 0.022), stenosis (r = 0.217, *p* = 0.002), and plaque burden (r = 0.283, *p* = 0.010).

**Table 1 diagnostics-13-00804-t001:** Baseline clinical data comparison between recurrence and non-recurrence group.

Index	Non-Recurrence Group (*n* = 158)	Recurrence Group (*n* = 41)	t/z/χ^2^	*p* Value
Age, years, mean ± SD	63.56 ± 12.98	67.10 ± 9.96	−1.59	0.113
Male, *n* (%)	102 (64.6)	32 (78.0)	2.69	0.134
Smoking history, *n* (%)	34 (21.5)	14 (34.1)	2.84	0.103
History of hypertension, *n* (%)	119 (75.3)	26 (63.4)	2.33	0.167
History of diabetes, *n* (%)	46 (29.1)	17 (41.5)	2.30	0.136
Glycosylated haemoglobin, mmol/L, mean ± SD	6.78 ± 1.50	7.15 ± 1.59	−1.33	0.186
Triglycerides, mmol/L, mean ± SD	1.67 ± 1.08	1.68 ± 0.94	−0.03	0.980
Total cholesterol, mmol/L, mean ± SD	3.97 ± 1.26	4.15 ± 1.07	−0.89	0.373
HDL, mmol/L, mean ± SD	1.20 ± 0.32	1.12 ± 0.30	1.37	0.172
LDL, mmol/L, mean ± SD	2.48 ± 1.16	3.13 ± 6.32	−0.62	0.537
Apolipoprotein a, mmol/L, mean ± SD	1.20 ± 0.23	1.15 ± 0.22	1.16	0.247
Apolipoprotein b, mmol/L, mean ± SD	0.93 ± 0.35	2.15 ± 17.87	−0.50	0.616
Hs-CRP, mmol/L, mean ± SD	5.54 ± 15.34	5.48 ± 16.47	0.02	0.980
Homocysteine, μmol/L, mean ± SD	39.83 ± 103.94	48.99 ± 143.81	−0.86	0.393
Cystatin C, mmol/L, mean ± SD	20.49 ± 110.57	38.05 ± 148.73	−0.77	0.443
sLOX-1, pg/mL, mean ± SD	936.36 ± 552.47	1351.68 ± 551.05	−4.29	<0.001

LDL = low-density lipoprotein; HDL = high-density lipoprotein; sLOX-1 = soluble lectin-like oxidised low-density lipoprotein receptor-1; SD, standard deviation; IQR, interquartile range.

**Table 2 diagnostics-13-00804-t002:** Comparison of HR-MR-VWI characteristics between recurrence and non-recurrence groups.

Characteristic	Non-Recurrence (*n* = 158)	Recurrence Group (*n* = 41)	t/z/χ^2^	*p* Value
Posterior circulation, *n* (%)	70 (44.3)	21 (51.2)	0.63	0.483
Plaque thickness, mm, mean ± SD	1.52 ± 0.40	1.70 ± 0.49	−2.19	0.033
Remaining lumen area, mm^2^, mean ± SD	4.40 ± 2.55	4.28 ± 3.49	0.25	0.804
Degree of stenosis, mean ± SD	41.14 ± 18.30	49.40 ± 21.15	−2.48	0.014
Plaque burden, median [IQR]	71 (65, 80)	76 (68, 85)	−2.57	0.010
Eccentric distribution, *n* (%)	97 (61.4)	20 (48.8)	2.14	0.157
Hyperintensity on T1WI, *n* (%)	43 (27.2)	27 (65.9)	21.31	<0.001
Positive Remodelling, *n* (%)	62 (39.2)	27 (65.9)	9.33	0.003
Significantly enhanced, *n* (%)	49 (31.0)	21 (51.2)	5.83	0.027

SD, standard deviation; IQR, interquartile range.

**Table 3 diagnostics-13-00804-t003:** Multivariate Cox regression of the significant characteristics associated with stroke recurrence.

Risk Factors	HR (95%CI)	*p* Value
sLOX-1, pg/mL	1.001 (1.000, 1.002)	0.002
Plaque thickness	1.515 (0.735, 3.120)	0.260
Degree of stenosis	1.006 (0.985, 1.028)	0.560
Plaque burden	1.003 (0.974, 1.032)	0.858
Hyperintensity on T1WI	2.326 (1.034, 5.231)	0.041
Positive Remodelling	1.282 (0.595, 2.763)	0.526
Significantly enhanced	0.871 (0.419, 1.812)	0.712

**Table 4 diagnostics-13-00804-t004:** Multivariate Cox regression of the significant characteristics associated with stroke recurrence.

Risk Factors	HR (95%CI)	*p* Value
sLOX-1 > 912.19 pg/mL	2.583 (1.142, 5.486)	0.023
Plaque thickness	1.342 (0.670, 2.689)	0.407
Degree of stenosis	1.007 (0.987, 1.028)	0.479
Plaque burden	0.999 (0.972, 1.026)	0.916
Hyperintensity on T1WI	2.632 (1.197, 5.790)	0.016
Positive Remodelling	1.316 (0.621, 2.791)	0.474
Significantly enhanced	0.914 (0.972, 1.026)	0.809

**Table 5 diagnostics-13-00804-t005:** Relationship between sLOX-1 levels and culprit plaque qualitative characteristics.

	sLOX-1	Q1	Q2	Q3	Q4	F	*p* Value
Characteristic							
Posterior circulation, *n* (%)		16 (32.0)	27 (54.0)	23 (46.9)	25 (50.0)	1.872	0.136
Eccentric distribution, *n* (%)		33 (66.0)	29 (58.0)	27 (55.1)	28 (56.0)	0.501	0.682
Hyperintensity on T1WI, *n* (%)		3 (6.0)	15 (30.0)	21 (42.9)	31 (62.0)	14.501	<0.001
Positive remodelling, *n* (%)		8 (9.0)	23 (25.8)	26 (29.2)	32 (36.0)	9.602	<0.001
Significantly enhanced, *n* (%)		6 (12.0)	16 (32.0)	21 (42.9)	27 (54.0)	7.684	<0.001

sLOX-1, soluble lectin-like oxidised low-density lipoprotein receptor-1; grouped according to the quartile of sLOX-1 levels: Q1 represents ≤ 591.92 pg/mL, Q2 represents 591.92–958.56 pg/mL, Q3 represents 958.56–1389.62 pg/mL, and Q4 represents ≥ 1389.62 pg/mL.

## Data Availability

Data is contained within the article. Data supporting the reported results may be provided upon reasonable request.

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
