# Peer review of "Correlation of sLOX-1 Levels and MR Characteristics of Culprit Plaques in Intracranial Arteries with Stroke Recurrence"

_diagnostics, 2023, doi:10.3390/diagnostics13040804_

Round 1

Reviewer 1 Report

The paper by Ren Et al, titled " Correlation of sLOX-1 levels and MR characteristics of culprit 2 plaques in intracranial arteries with stroke recurrence “ was an interesting read.

Authors have demonstrated that there was a significant association with the level of sLOX-1 and recurrent acute ischemic stroke. However, I have a major concern with this. The authors had reported sLOX-1 level as HR of 1.001 with a very narrow 95% CI 1.000-1.002 giving a p value of 0.002 does not make sense based on the data provided in other sections of the article. Please provide and explanation and also the raw data of the sLOX-1 levels used to calculate this.

Other minor revisions needed are

Line 11 – completely expand the abbreviation

Line 82 – what baseline data were needed to be said as ‘complete’, this is not a proper inclusion criteria

Line 93 – correction - dual antiplatelet treatment

Line 144 – clarify the definition for negative remodelling, it appears to be incorrect

Line 154 – grammar correction is needed

Line 355 – burdenof – missing space.

Author Response

Dear Professor:

Thank you for your comments concerning our manuscript entitled “Correlation of sLOX-1 levels and MR characteristics of culprit plaques in intracranial arteries with stroke recurrence” (ID: diagnostics-2125264). Those comments are all valuable and very helpful for revising and improving our paper, as well as the important guiding significance to our research. We have studied comments carefully and have made supplement which we hope meet with approval. The main correction in the paper and the responds to your comments are as the word file below:

Reviewer 2 Report

The manuscript by Ren et al. shows the importance of sLOX-1 level in predicting of stroke recurrence. Furthermore, there are correlations with vessel MRI parameters.

Overall, the authors provide interesting and important data for the field of stroke, although there have already been some reports on the importance of sLOX-1 in stroke.

Some aspects of the manuscript have to be reconsidered:

1)       The abstract is too long. It would be enough if some detailed information appears in the later text.

2)     Why is it necessory to use dual antiplatelet therapy in all patients?

3)     For better understanding, please explain figure 3 and Table 4 in the text in more detail.

Author Response

(The authors gave the same response as above.)

Round 2

Reviewer 1 Report

Good work. I would recommend adding the hazard ratio and p value for sLOX-1 when analyzed as a classification / categorical - binary variable as reported in the revision word file. 

Author Response

Dear Professor:

Thank you for your recommendation concerning our manuscript entitled “Correlation of sLOX-1 levels and MR characteristics of culprit plaques in intracranial arteries with stroke recurrence” (ID: diagnostics-2125264). The recommendation is valuable and very helpful for revising and improving our paper, as well as the important guiding significance to our research. We have revised the manuscript according to your recommendation.

Reviewer 2 Report

No further concerns on my side!

Round 3

Reviewer 1 Report

Looks better.